# Cross-coupling polycondensation via C–O or C–N bond cleavage

Ze-Kun Yang[1,2], Ning-Xin Xu[1], Ryo Takita [2], Atsuya Muranaka[2], Chao Wang [1,2] & Masanobu Uchiyama [1,2]

π-Conjugated polymers are widely used in optoelectronics for fabrication of organic photovoltaic devices, organic light-emitting diodes, organic field effect transistors, and so on. Here we describe the protocol for polycondensation of bifunctional aryl ethers or aryl ammonium salts with aromatic dimetallic compounds through cleavage of inert C–O/C–N bonds. This reaction proceeds smoothly in the presence of commercially available Ni/Pd catalyst under mild conditions, affording the corresponding π-conjugated polymers with high molecular weight. The method is applicable to monomers that are unreactive in other currently employed polymerization procedures, and opens up the possibility of transforming a range of naturally abundant chemicals into useful functional compounds/polymers.

[1] Graduate School of Pharmaceutical Sciences, The University of Tokyo, 7-3-1 Hongo, Bunkyo-ku, Tokyo 113-0033, Japan. [2] Cluster of Pioneering Research (CPR), Advanced Elements Chemistry Laboratory, RIKEN, 2-1 Hirosawa, Wako, Saitama 351-0198, Japan. Correspondence and requests for materials should be addressed to C.W. (email: chaowang@mol.f.u-tokyo.ac.jp) or to M.U. (email: uchiyama@mol.f.u-tokyo.ac.jp)

$\pi$-Conjugated polymers including poly(arene)s are important materials in the field of optoelectronics, e.g., for fabricating organic photovoltaic devices (OPVs)[1,2], organic light-emitting diodes (OLEDs)[3], and organic field effect transistors (OFETs)[4]. Conventionally, these polymers have been synthesized by polycondensation based on transition-metal catalyzed cross-coupling reactions, such as Kumada–Tamao–Corriu coupling[5,6], Suzuki–Miyaura coupling[6,7], Stille coupling[8], and so on[9,10]. Most existing transition-metal-catalyzed polycondensations affording $\pi$-conjugated polymers require dihaloarenes as monomers, although the development of direct C–H arylation polymerization (DArP) has recently greatly expanded the substrate scope[11].

On the other hand, aromatic C–O/C–N bonds are ubiquitous in natural products, pharmaceuticals, and functional molecules, and many compounds containing these bonds, such as ethers and anilines, are produced on an industrial scale at reasonable cost. In recent years, synthetic transformations via inert aromatic C–O[12–17] and/or C–N[18] bond cleavage have attracted a great deal of attention, not only as an alternative to the use of reactive aromatic halides, but also as a new methodology for the direct conversion and rapid derivatization of functional molecules containing aromatic C–O/C–N bonds. The first Ni-catalyzed C–O cleavage cross-coupling reaction of aryl methyl ethers with Grignard reagents was reported by Wenkert et al. as early as in 1979[19,20]. However, it was then largely neglected for almost 30 years until Dankwardt significantly extended the substrate scope[21]. Since then, several groups, including ours, have developed various types of cross-coupling reactions of aryl ethers, including Kumada–Tamao–Corriu type (Mg)[22–28], Suzuki–Miyaura type (B)[29–35], Negishi type (Zn, Al)[36–39], Murahashi type (Li)[40–43], and related C–H[44–50] or C–X (X = B, N, Si, etc.)[51–55] bond formation reactions. However, it remains unclear whether such inert ethereal C–O bond cleavage can be extended to synthesize functional polymers.

Herein, we report polycondensation using aromatic ether or ammonium salt as a monomer for the synthesis of various $\pi$-conjugated polymers via inert C–O/C–N bond cleavage (Fig. 1).

## Results

**Polycondensation of organometallic reagent with aryl ether.** Considering the high reactivity and wide availability of Grignard reagents, we chose the Kumada–Tamao–Corriu type reaction for the initial test, with Grignard reagent **1a** and 2,6-dimethoxynaphthalene **2a** as model reactants. The fluorene moiety is an important structural unit in various functional $\pi$-conjugated polymers. $NiCl_2(PCy_3)_2$ ($PCy_3$: tricyclohexylphosphine) was selected as the catalyst, as it is known to be effective for ethereal C–O bond cleavage[12–17,21–28]. Grignard reagent **1a** was initially prepared by treating 2,7-dibromo-9,9-dihexylfluorene (**0a**) with magnesium chips in refluxing THF (tetrahydrofuran) (Method A, Table 1). Gratifyingly, an initial attempt in toluene at 120 °C afforded the desired polycondensation product **3aa** with $M_n$ (number-average molecular weight) of 5.3 kDa and PDI (polydispersity index) of 1.80. Examination of several solvents showed toluene to be greatly superior to etheric solvents (Entries 1–3). During optimization of the reaction conditions, we found that the quality/purity of **1a** critically influenced the yield, reactivity, and behavior of this polycondensation. When **1a** freshly prepared according to Method A was quenched with iodine, a mixture of the *di*- and *mono*-iodides (9,9-dihexyl-2,7-diiodofluorene and 9,9-dihexyl-2-iodofluorene) was obtained, probably due to partial degradation of the C–Mg moiety by THF at the high reaction temperature. We speculated that *mono*-Grignard reagents presumably terminate the chain growth and reduce the molecular weight of the products. Thus, we examined preparation methods for **1a**. Addition of 0.01 equivalent of 1,2-dibromoethane (as an initiator) to the reaction mixture of **0a** and Mg turnings was effective; the formation of **1a** was greatly accelerated and was completed within 1 h at room temperature (Method B), resulting in dramatically decreased formation of *mono*-Grignard by-product and smooth polycondensation to give **3aa** ($M_n = 9.7$ kDa) (Entry 4). Lastly, we examined Rieke's method[56] for the preparation of active magnesium (Mg*), which enabled rapid insertion of Mg to the C–Br bond at very low temperature (Method C). Polycondensation using the resultant Grignard reagent **1a** proceeded very cleanly to afford high-molecular weight polymer **3aa** ($M_n = 11.6$ kDa) (Entry 5). The polymerization temperature also played a key role in determining the yield and $M_n$ of this polycondensation (Entries 6 and 7), and a higher molecular weight ($M_n = 21.1$ kDa) of the product **3aa** were obtained at room temperature (Entry 7). A higher catalyst loading (10 mol%) did not dramatically change the outcome, while a lower catalyst loading (2.5 mol%) gave **3aa** with moderate molecular weight ($M_n = 7.7$ kDa). It is known that lithium salts can formate complexes with several organometallics, such as Grignard reagents, and this can enhance the reactivity in some chemical transformations[57,58]. However, addition of LiCl to the current reaction system did not change the outcome (LiCl was first added to the THF solution of **1a**, which was stored for hours to allow the complexation to occur[59]).

We next turned our attention to other organometallic species. Recently, Rueping,[40] Feringa,[41] and our group[42–43] have reported protocols for ethereal Murahashi-type coupling using organolithiums, which provide high efficiency under mild conditions. When di-lithium reagent **4a** prepared by the bromine-lithium exchange reaction of **0a** was employed, the polycondensation with **2a** took place smoothly at room temperature with Ni(cod)2 catalyst (cod: 1,5-cyclooctadiene) and SIMes ligand (SIMes: 1,3-bis(2,4,6-trimethylphenyl) imidazolidin-2-ylidene), affording **3aa** in 80% yield with high molecular weight ($M_n = 15.3$ kDa, PDI = 1.92) (Table 1, Entry 8). On the other hand, polycondensation of **2a** with organoboron reagent[29–35] as well as organozincate[36] gave unsatisfactory results, yielding low-molecular-weight products ($M_n < 3.0$ kDa) under various conditions. Finally, the set of conditions shown in Entry 7, Table 1 was found to be optimal in terms of efficiency and molecular weight.

Under the optimal conditions, we next examined the scope of the present polycondensation for various aromatic ethers **2**, with Grignard reagent **1a/1b**. The results are summarized in Fig. 2. We

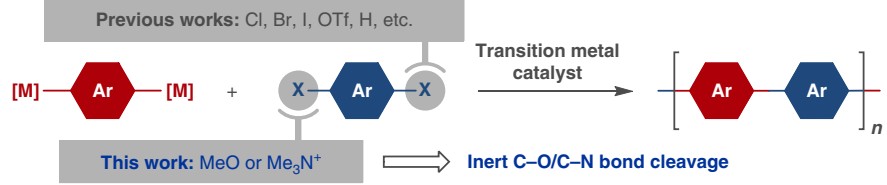

**Fig. 1** Brief background. Conventional methods/electrophiles for cross-coupling polycondensation, and the C–O/C–N bond-cleavage protocol developed in this work

**Table 1 Screening of reaction conditions for Ni-catalyzed cross-coupling polycondensation between Grignard reagent 1a and 2,6-dimethoxynaphthalene 2a**

| Entry | Synthesis of 1a | T (°C) | Solvent[a] | $M_n$ (kDa) | PDI | Yield (%)[b] |
|---|---|---|---|---|---|---|
| 1 | Method A | 120 | Toluene | 5.3 | 1.80 | not isolated[c] |
| 2 | Method A | 120 | $^nBu_2O$ | 2.1 | 2.01 | not isolated[c] |
| 3 | Method A | 120 | $^iPr_2O$ | 1.5 | 1.34 | not isolated[c] |
| 4 | Method B | 120 | Toluene | 9.7 | 2.52 | not isolated[c] |
| 5 | Method C | 120 | Toluene | 11.6 | 2.64 | 84 |
| 6 | Method C | 70 | Toluene | 11.9 | 2.36 | 88 |
| 7 | Method C | 25 | Toluene | 21.1 | 2.90 | 93 |
| 8 | (structure 4a) | 25 | Toluene / Ni(cod)$_2$ (10.0 mol%), SIMes (10.0 mol%) | 15.3 | 1.92 | 80 |

[a] For all reactions, THF in the solution of **1a** was removed under vacuum at 0 °C before adding the indicated solvent
[b] Yields were calculated after isolation (precipitation in MeOH)
[c] The products showed little or no precipitation in MeOH

found that **1a** freshly prepared from Rieke Mg* and **2a** underwent polycondensation smoothly to give the corresponding polymer **3aa** with high molecular weight ($M_n = 23.2$ kDa, PDI = 3.02, in 0.3 M solution) in 96% isolated yield. 2,7-Dimethoxynaphthalene **2b** and 1,6-dimethoxynaphthalene **2c** were also good substrates, and the polycondensations with **1a** at room temperature gave **3ab** ($M_n = 23.6$ kDa, PDI = 2.73) and **3ac** ($M_n = 13.2$ kDa, PDI = 2.53) in 96% and 82% yields, respectively. The reaction of 1,4-dimethoxynaphthalene **2d** bearing C–O bonds at the α-position required the use of NHC (N-heterocyclic carbene) ligand (ICy:[26,32,35] 1,3-bis(2,6-diisopropylphenyl)imidazolidin-2-ylidene) at a higher reaction temperature, giving **3ad** in 83% yield ($M_n = 11.3$ kDa, PDI = 2.28). It is known that Ni-catalyzed cross-coupling via inert C–O bond cleavage sometimes suffers from low reactivity toward phenolic substrates[12–17]. Indeed, polycondensation using 1,4-dimethoxybenzene **2e** and **1a** with Ni(PCy$_3$)$_2$Cl$_2$ catalyst proceeded sluggishly even at higher temperature, affording lower-molecular-weight **3ae** (at r.t.: $M_n = 3.5$ kDa, at 70 °C: $M_n = 4.9$ kDa). A big improvement was observed when ICy ligand was used in the polycondensation between **1a** and **2e** at 70 °C, providing **3ae** in 88% yield ($M_n = 12.1$ kDa, PDI = 2.32). NHC ligands have been used for Ni-catalyzed C–O bond cleavage reactions, and show higher activity than PCy$_3$ in some cases, probably because they can stabilize the transition state for oxidative addition of C–O bonds to Ni[35]. Hence, we think the use of NHC ligand probably accelerated the C–O bond cleavage step, favouring the polycondensation as compared with the decomposition of the Grignard reagent, and thus leading to a greater yield and a higher molecular weight. Other anisole derivatives such as **2f**, **2g**, and **2h** also reacted without difficulty under the same reaction conditions, affording the corresponding polymers **3af–3ah** in high yields ($M_n = 12.7–14.3$ kDa, PDI = 2.33–2.56). Further, a chemo-selective polycondensation of **2i** was achieved even in the presence of the double bond, giving polymer **3ai** with the gem-sp$^2$ linker in 82% yield ($M_n = 9.2$ kDa, PDI = 2.11). Under the current conditions, the reaction of 2,8-dimethoxydibenthiophene was rather sluggish, but 2,8-dimethoxydibenzofuran **2j** reacted with **1a** without difficulty, and the desired polymer **3aj** was obtained in 93% yield ($M_n = 10.6$ kDa, PDI = 2.29). This reaction is not limited to fluorene-tethered Grignard reagent **1a**, and a carbazole derivative **1b** afforded the corresponding product **3ba** in 92% yield ($M_n = 17.9$ kDa, PDI = 2.58). It should be noted that many of these aryl ether monomers, such as **2a-g**, are commercially available at reasonable cost (according to SciFinder), while the corresponding di-bromides or di-iodides are either more expensive or less readily accessible. Hence, the current ethereal polycondensation should be a useful supplement to the well-established polymerization protocols based on the use of dihaloarenes.

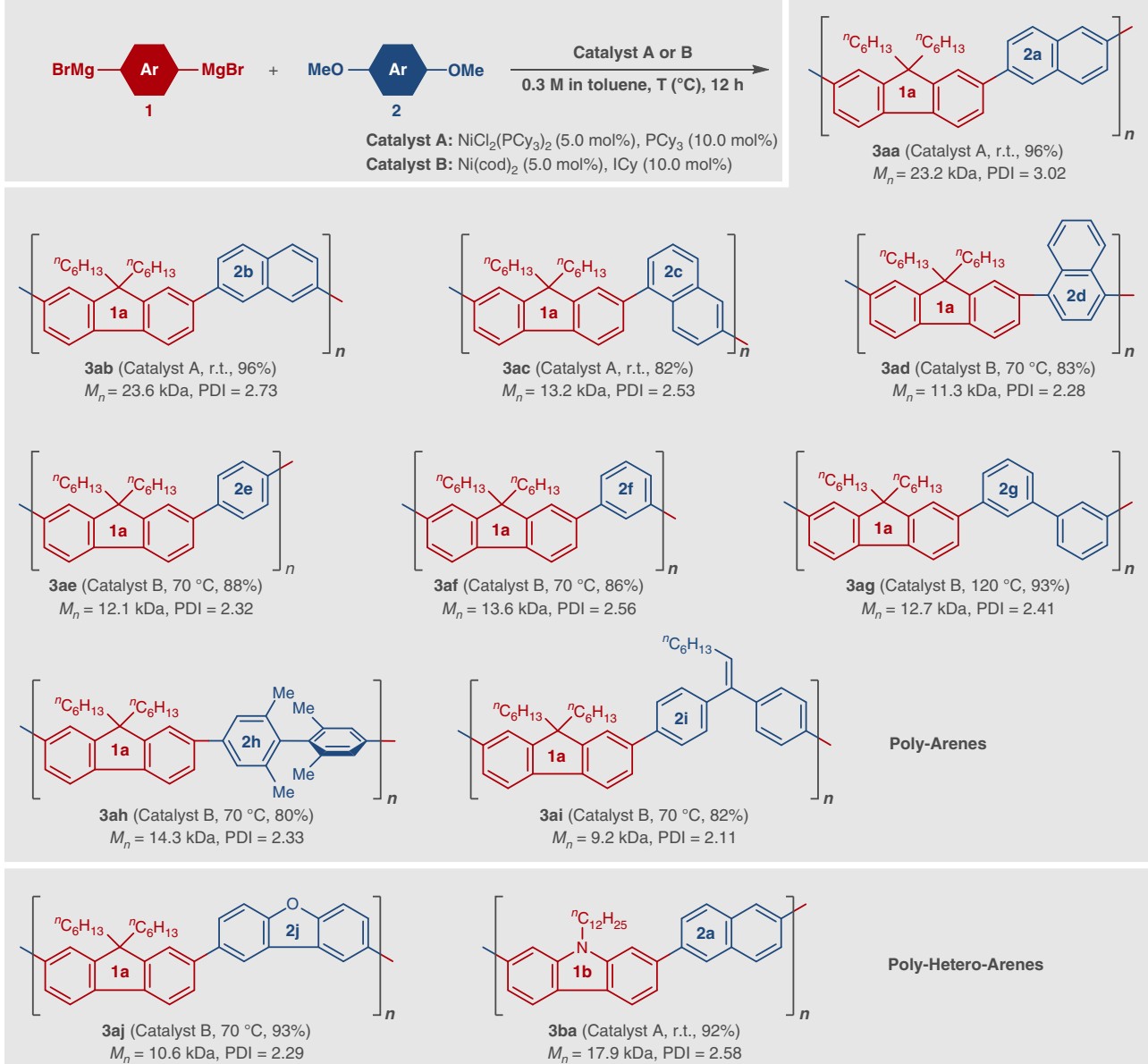

**Fig. 2** Reaction Scope (1). Ni-catalyzed cross-coupling polycondensation between Grignard reagents **1** and dimethoxyarenes **2**

Axial asymmetric structures are widely utilized in functional molecules, asymmetric synthesis and related areas. For instance, BINOL (2,2′-dihydroxy-1,1′-binaphthyl) derivatives have been extensively used as catalysts or ligands for asymmetric synthesis, and also are important motifs in supramolecular architectures and components of optical materials[60]. Thus, we examined whether the BINOL unit can be installed in the polymers obtained with the present polycondensation strategy. However, all attempts at preparing the intermediary Grignard reagent having a BINOL skeleton under various conditions were unsuccessful, probably due to low reactivity of the corresponding bromides. On the other hand, di-lithium reagents **4c**[R] or **4c**[S] could be prepared easily via a Li/Br exchange protocol (Supplementary Fig. 1). After extensive experimentation, we found that Murahashi-type polycondensation of **2a** with **4c** was effective, providing **3ca** in high yield (Fig. 3). The ethereal moieties on the BINOL ring remained intact, indicating that the reaction could be used for chemoselective polycondensation of substrates having multiple C–O bonds. Further, no racemization occurred at the axially

asymmetric positions (Supplementary Fig. 2). These results demonstrate the potential applicability of this method to generate various axially asymmetric architectures of functional molecules/materials, which may have unusual optical properties.

**Polycondensation of grignard reagent with ammonium salt.** With the ethereal C–O bond-cleavage polycondensation in hand, we next considered whether the protocol could also be applied to C–N bond cleavage. Amine groups occur in many biologically essential compounds and are widely used in industry for preparation of pharmaceuticals and functional molecules, but the high stability of amino C–N bonds makes them highly resistant to cleavage reactions[18]. On the other hand, quaternary aryl ammonium salts can be easily prepared from various anilines, and such C–N bonds exhibit higher reactivity than those of amines. Following the pioneering work by Wenkert et al. in 1988[61], applications of aryl ammonium salts for cross-coupling and related reactions have been rapidly developed in recent years

**Fig. 3** Reaction Scope (2). Ni-Catalyzed polycondensation of **2a** with organolithium **4c^R** or **4c^S** having BINOL skeletons

**Fig. 4** Reaction Scope (3). Pd-catalyzed cross-coupling polycondensation between Grignard reagent **1** and ammonium salt **5**

by several groups[62–70] including ours[39,42,69,70]. Here, we first tried polycondensation using **1a** and **5k** or **5l**, with two ammonium moieties (Fig. 4). The reaction proceeded smoothly in THF at room temperature with 1 mol% PdCl₂(PPh₃)₂[64], giving **3ak** and **3al** in 95% ($M_n = 23.4$ kDa, PDI = 2.64) and 88% yields ($M_n = 24.0$ kDa, PDI = 2.51), respectively. Interestingly, the reaction of **5k** with Grignard reagent **1d** having dibenzothiophene was also successful, and **3dk** was obtained in 84% yield ($M_n = 13.5$ kDa, PDI = 2.48). (Benzo)thiophene units have been incorporated into many π-conjugated polymers, due to their excellent photoelectrical properties, and the current method provides a route for syntheses of such molecules.

## Discussion

In conclusion, we have developed an effective synthetic method for π-conjugated polymers from various aromatic ethers or ammonium salts through C–O or C–N bond cleavage. With commercially available Ni or Pd catalysts, cross-coupling polycondensation between Grignard reagents/organolithiums and ethers/ammoniums took place smoothly under mild conditions, providing the polycondensation products with high molecular weight in good to excellent yield. The purity of the organometallic coupling partners is important for achieving a smooth reaction. The current results not only provide a methodology for inert C–O/C–N bond cleavage cross-coupling reactions, but also open up a new route for the synthesis of various functional molecules/polymers. Further investigations of the reaction scope (e.g., organoboron, organozinc, and/or other C–O/C–N electrophiles) and synthetic applications of the present method to photoelectrical materials, are in progress. It should be noted that previous mechanistic studies on transition metal-catalyzed C–O/C–N bond cleavage reactions from our group[27,35,69] and others[35,43,46,47,49] have found that the oxidative addition of ethereal C–O bond to Ni generally suffers from high activation barriers. However, it can be greatly facilitated either by formation of the Ni(0)-ate complex with Grignard reagent or by NHC ligand. On the other hand, due to the high reactivity, C–N bond cleavage in aryl

ammonium salts with transition metals is kinetically and thermodynamically favorable, and the subsequent transmetalation becomes rate-determining. In this context, we are conducting a comprehensive mechanistic study of the present polycondensation reactions using both theoretical and experimental methods, aiming to improve the reactivity and selectivity, as well as to provide a sound basis to design and establish new polymerization protocols.

## Methods

**Typical procedure for polycondensation.** To a dry Schlenk flask charged with argon, Grignard reagent **1** (0.3 mmol, THF solution) was added. THF was then removed under vacuum at 0 °C, and toluene (1 mL) was added. The mixture was stirred for 5 min and evaporated again under vacuum. Next, dimethoxyarenes **2** (0.3 mmol), Catalyst A [NiCl₂(PCy₃)₂ (5 mol%) and PCy₃ (10 mol%)] or B [Ni (cod)₂ (5 mol%) and ICy (10 mol%, prepared in situ by treating ICy • HCl with stoichiometric amounts of EtMgBr)] and toluene (1 mL) were added. This reaction mixture was stirred overnight (>12 h) at room temperature or with heating, and then quenched with 1 M HCl (5 mL). The aqueous layer was extracted with CH₂Cl₂ (3 × 5 mL), and the organic solution was dried over MgSO₄, filtered, and concentrated under vacuum. The residue was dissolved in a minimum amount of CH₂Cl₂ (ca. 1 mL) and precipitated by adding the solution to MeOH (100 mL). The precipitate was collected and dried under vacuum. For full details, see Supplementary Figs. 1–18, Supplementary Tables 1–3 and Supplementary Methods

**Data availability.** Detailed experimental procedures and characterization of compounds can be found in the Supplementary Information (Supplementary Figs. 1–18, Supplementary Tables 1–3 and Supplementary Methods). All data are available from the authors on reasonable request.

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

## Acknowledgements

This work was supported by JSPS Grant-in-Aid for Scientific Research on Innovative Areas (No. 17H05430) (to M. U.), JSPS KAKENHI (S) (No. 17H06173) (to M. U.), JSPS KAKENHI (C) (No. 18K06544) (to C. W.), and by grants from Kobayashi International Scholarship Foundation (to M. U. and C. W.), and YakuGaku Shin-KoKai Foundation (to C. W.). Z.-K.Y. is grateful for the Junior Research Associate fellowship provided by RIKEN and the JSPS Research Fellowships for Young Scientists (DC1).

## Author contributions

Z.-K.Y. planned, conducted, analysed, and summarized the experiments. N.-X.X. performed some experiments. A.M. and R.T. participated in discussions. C.W. and M.U. conceived, designed, and supervised the project. Z.-K.Y., C.W., and M.U wrote the manuscript with feedback from N.-X.X., A.M., and R.T.

## Additional information

**Competing interests:** The authors declare no competing interests.

