## [Peer Review File · Nature Communications]

Reviewers' comments:

Reviewer #1 (Remarks to the Author):

This manuscript by Uchiyama and Wang reports elegant work on novel Ni or Pd catalysed polycondensation reactions by the cleavage of relatively inert C-O and C-N bonds in aromatic scaffolds. Using di-Mg and di-lithium organometallic precursors, the authors demonstrate the scope of this method to access several pi-conjugated polymers by cross-coupling with a range of aromatic ethers and ammonium salts. Reactions take place under mild conditions using commercially available Ni and Pd catalysts, which should encourage other synthetic and materials chemists to adopt this approach for the preparation of valuable supramolecules that have many applications in OLEDs and OPVs. Thus from the point of view of synthesis and applicability, as well as the novelty of this method, I give this paper a positive recommendation for publication.

Having said that, I find there are a few gaps in the knowledge that the authors should address in the future, although this may be better suited for a future full paper rather than an in depth revision of the current manuscript.

Reactions in Table 1 seem to be strongly dependent on the donor ability of the solvent. Toluene seems the best option, however the Grignard precursors are prepared in THF. For clarity the main text or table 1 should state that the bulk of THF is removed before the coupling reaction takes place. Authors also mention LiCl did not affect the outcome of the reaction, but was this reaction attempted in toluene? If so it is unlikely that LiCl will dissolve under the conditions of the study and hence I would be cautious on any conclusion drawn from this experiment.

For the condensation of 1a and 2e, a significant improvement is observed by introducing an NHC as a coligand. A similar co-catalyst is used for the couplings using dilithio derivatives 4a and 4c. It is not obvious to this reviewer the role that this co-ligand plays in these reactions and how it contributes towards increasing the selectivity of the coupling process. This needs some elaboration.

While mechanistic investigations are underway, I feel the reader will greatly benefit if the authors could at least comment on how these couplings are likely to occur, even if it is relating their findings to previous mechanistic studies in Ni-catalysed C-O bond activation reported by others such as Chatani or Martin.

Reviewer #2 (Remarks to the Author):

This manuscript by Uchiyama and coworkers describes their design of a new metal method to access conjugated polyaromatics from arylothers and arylamines. While the use of Pd and Ni catalyzed cross-coupling reactions remains the most commonly employed approach to these classes of polymers, expanding upon these platforms to new classes of non-halogenated and/or metallated monomers is an important goal in creating more streamlined and less wasteful routes to these polymers. Arene C-H functionalization provides perhaps the most general direct alternative to cross coupling. However, the work here by Uchiyama is intriguing as an alternative, as aryl-O and aryl-N monomers are in some cases broadly available. In this regard, one issue that could be better addressed is the accessibility of some of the specific aryldiether monomers used in this chemistry, and how that compares to aryl halides? I would assume this would be straightforward for some of the substrates employed (e.g. hydroquinone derivatives). Assuming this and related issues below can be addressed, I believe this would be a nice contribution to Nature Communications.

Other comments:

1. Table 1: the polymers yields is missing from the table. These would presumably significantly affect molecular weights in each entry, and should be added.
2. Figure 2: is there a reason why heterocyclic diethers (e.g. furan, thiophene) were not probed in the diversity table? As these are common monomers in conjugated polymers, they would be worth a comment/footnote if they are not viable under these conditions.
3. Figure 3: I am unclear on why this binol conjugated polymer merits its own discussion in the paper? Is there a useful expected property of the conjugated polymer? If not, this discussion could be streamlined.
4. Line 60: simplify to, "Mono-Grignard reagents presumably..." Line 118: "Axial asymmetric structures are..."

Reviewer #3 (Remarks to the Author):

This cross-coupling method described in this manuscript is a new and very useful variation of the Kumada cross-coupling procedure. It allows the elaboration of complex polymers using simple aromatic methoxy-substituted aromatic building blocks. The scope of this polycondensation procedure is good and high molecular weights of very useful polymers are obtained. This new polymer will be of interest to a broad range of chemists. The manuscript is very well written and may open new synthetic possibilities in the construction of macromolecules. It will certainly stimulate the design of new type of polymers.

Paul Knochel

Response to Reviewers' comments:

Reviewer 1

Comment 1-1: This manuscript by Uchiyama and Wang reports elegant work on novel Ni or Pd catalysed polycondensation reactions by the cleavage of relatively inert C-O and C-N bonds in aromatic scaffolds. Using di-Mg and di-lithium organometallic precursors, the authors demonstrate the scope of this method to access several pi-conjugated polymers by cross-coupling with a range of aromatic ethers and ammonium salts. Reactions take place under mild conditions using commercially available Ni and Pd catalysts, which should encourage other synthetic and materials chemists to adopt this approach for the preparation of valuable supramolecules that have many applications in OLEDs and OPVs. Thus from the point of view of synthesis and applicability, as well as the novelty of this method, I give this paper a positive recommendation for publication.

Response: Thank you for your high evaluation of this work.

Comment 1-2: Having said that, I find there are a few gaps in the knowledge that the authors should address in the future, although this may be better suited for a future full paper rather than an in depth revision of the current manuscript.

Reactions in Table 1 seem to be strongly dependent on the donor ability of the solvent. Toluene seems the best option, however the Grignard precursors are prepared in THF. For clarity the main text or table 1 should state that the bulk of THF is removed before the coupling reaction takes place. Authors also mention LiCl did not affect the outcome of the reaction, but was this reaction attempted in toluene? If so it is unlikely that LiCl will dissolve under the conditions of the study and hence I would be cautious on any conclusion drawn from this experiment.

Response: Thank you for your important comments. First, as you suggested, we have clearly stated in Table 1 that THF in the solution of **1a** was removed under vacuum at 0 °C before adding toluene for the next cross-coupling. Second, in our protocol, LiCl was added in the THF solution of Grignard reagent and the mixture was kept for hours to complete the complexation of RMgX-LiCl. As reported, the complex is stable and could be crystallized from THF. We have included these points in the revised manuscript and have added corresponding literature in the reference list.

We appreciate your encouragement regarding a potential future full paper - we are of course continuing work to extend the scope of the reaction and establish its limitations, and certainly hope to publish the results in detail in due course.

Comment 1-3: For the condensation of **1a** and **2e**, a significant improvement is observed by introducing an NHC as a coligand. A similar co-catalyst is used for the couplings using dilithio derivatives **4a** and **4c**. It is not obvious to this reviewer the role that this co-ligand plays in these reactions and how it contributes towards increasing the selectivity of the coupling process. This needs some elaboration.

Response: Thank you for your constructive suggestion. In recent years, NHC ligands have been found to improve the reactivity/selectivity of some ethereal C–O bond cleavage reactions. Very recently, Mori, Chatani and Tobisu presented a detailed mechanistic study on the Ni/NHC-mediated C–O bond cleavage. According to them, NHC ligands effectively stabilize the transition state of the oxidative addition step and accelerate the C–O bond cleavage. We think that in our case, the use of the NHC ligand probably accelerates the C–O bond cleavage step, and therefore polycondensation would be favoured over decomposition of the Grignard reagent, resulting in a better yield and a higher molecular weight. As you suggested, we have added a brief discussion and also have cited the above reference.

Comment 1-4: While mechanistic investigations are underway, I feel the reader will greatly benefit if the authors could at least comment on how these couplings are likely to occur, even if it is relating their findings to previous mechanistic studies in Ni-catalysed C-O bond activation reported by others such as Chatani or Martin.

Response: Thank you for your constructive suggestion. We have added a brief comment on the mechanistic issue to help readers to understand this chemistry.

Reviewer 2

Comment 2-1: This manuscript by Uchiyama and coworkers describes their design of a new metal method to access conjugated polyaromatics from arylothers and arylamines. While the use of Pd and Ni catalyzed cross-coupling reactions remains the most commonly employed approach to these classes of polymers, expanding upon these platforms to new classes of non-halogenated and/or metallated monomers is an important goal in creating more streamlined and less wasteful routes to these polymers. Arene C-H functionalization provides perhaps the most general direct alternative to cross coupling. However, the work here by Uchiyama is intriguing as an alternative, as aryl-O and aryl-N monomers are in some cases broadly available. In this regard, one issue that could be better addressed is the accessibility of some of the specific aryldiether monomers used in this chemistry, and how that compares to aryl halides? I would assume this would be straightforward for some of the substrates employed (e.g. hydroquinone derivatives). Assuming this and related issues below can be addressed, I believe this would be a nice contribution to Nature Communications.

Response: Thank you for your high evaluation of this work. Regarding accessibility, a search of SciFinder showed that many of these aryl ether monomers, such as **2a-g**, are commercially available at reasonable cost, whereas the corresponding di-bromides or di-iodides are either more expensive or less readily accessible. As you suggested, we have added a comment on this in the text. This supports the idea that the current ethereal polycondensation would be a useful supplement to the well-established polymerization protocols based on the use of dihaloarenes.

Comment 2-2: Other comments:

1. Table 1: the polymers yields is missing from the table. These would presumably significantly affect molecular weights in each entry, and should be added.

2. Figure 2: is there a reason why heterocyclic diethers (e.g. furan, thiophene) were not probed in the diversity table? As these are common monomers in conjugated polymers, they would be worth a comment/footnote if they are not viable under these conditions.

3. Figure 3: I am unclear on why this binol conjugated polymer merits its own discussion in the paper? Is there a useful expected property of the conjugated polymer? If not, this discussion could be streamlined.

4. Line 60: simplify to, "Mono-Grignard reagents presumably..." Line 118: "Axial asymmetric structures are..."

Response: Thank you for your constructive suggestions. We have made appropriate revisions, including:

1) We have added the yields of polymer products obtained by using Rieke-Grignard reagents (method C) in Table 1. In the reactions by method A or B, no polymers (or only trace amounts) precipitated in MeOH, due to the low molecular weights.

2) We performed the reactions with heterocyclic diethers as you suggested and the results have been added in the revised manuscript. The monomer **2j** containing a furan moiety reacted smoothly with **1a**, giving **3aj** in 93% yield ($M_n = 10.6$ kDa, PDI = 2.29). However, when 2,8-dimethoxydibenzothiophene was used, the reaction became very sluggish. Alternatively, a thiophene moiety could be introduced through the reaction between Grignard reagent **1d** and ammonium salt **5j** in high yield and molecular weight.

3) BINOL derivatives are important motifs in supra-molecular architectures and are also components of optical materials. The axial chirality provides interesting optical properties, such as circularly polarized luminescence (CPL). In the reaction between di-ether **2a** and organolithium reagent **4c** containing BINOL skeleton, no racemization occurred at the axially asymmetric positions, as reflected in the CD spectrum (Supplementary Fig. 2). We should prefer to retain this section to convey the scope of this methodology for axially chiral monomers, as well as the potential applications, to the broad readership of *Nature Communications*. However, we are willing to shorten or modify it, if you feel it is necessary.

4) We have revised the sentences according to your suggestions.

Reviewer 3

Comment 3-1: This cross-coupling method described in this manuscript is a new and very useful variation of the Kumada cross-coupling procedure. It allows the elaboration of complex polymers using simple aromatic methoxy-substituted aromatic building blocks. The scope of this polycondensation procedure is good and high molecular weights of very useful polymers are obtained. This new polymer will be of interest to a broad range of chemists. The manuscript is very well written and may open new synthetic possibilities in the construction of macromolecules. It will certainly stimulate the design of new type of polymers

Response: Thank you for your high evaluation of this work.

REVIEWERS' COMMENTS:

Reviewer #1 (Remarks to the Author):

Authors have made an excellent job in addressing the comments from the reviewers. Accordingly I am happy to recommend the acceptance of this manuscript in its present form.

Reviewer #2 (Remarks to the Author):

The revised manuscript addresses all of my previous concerns. I would recommend its acceptance with one small addition.

lines 130 and 140: references to both of these uses should be provided in these sentences noted.

RESPONSE TO REVIEWERS' COMMENTS:

Reviewer #1 (Remarks to the Author):

Authors have made an excellent job in addressing the comments from the reviewers. Accordingly I am happy to recommend the acceptance of this manuscript in its present form.

Reviewer #2 (Remarks to the Author):

The revised manuscript addresses all of my previous concerns. I would recommend its acceptance with one small addition.

lines 130 and 140: references to both of these uses should be provided in these sentences noted.

Response: As Reviewer 2 suggested ("lines 130 and 140: references to both of these uses should be provided in these sentences noted"), we have added the corresponding reference (ref. 60).